# Immunoglobulin G and Complement as Major Players in the Neurodegeneration of Multiple Sclerosis

**DOI:** 10.3390/biom14101210

**Published:** 2024-09-26

**Authors:** Peter G. E. Kennedy, Matthew Fultz, Jeremiah Phares, Xiaoli Yu

**Affiliations:** 1Institute of Neuroscience and Psychology, University of Glasgow, Glasgow G61 1QH, UK; peter.kennedy@glasgow.ac.uk; 2Department of Neurosurgery, University of Colorado Anschutz Medical Campus, Aurora, CO 80045, USA; matthew.fultz@cuanschutz.edu (M.F.); jeremiah.phares@cuanschutz.edu (J.P.)

**Keywords:** multiple sclerosis, neurodegeneration, IgG, complement, neurons, disease progression

## Abstract

Multiple Sclerosis (MS) is an inflammatory, demyelinating, and neurodegenerative disease of the central nervous system (CNS) and is termed as one of the most common causes of neurological disability in young adults. Axonal loss and neuronal cell damage are the primary causes of disease progression and disability. Yet, little is known about the mechanism of neurodegeneration in the disease, a limitation that impairs the development of more effective treatments for progressive MS. MS is characterized by the presence of oligoclonal bands and raised levels of immunoglobulins in the CNS. The role of complement in the demyelinating process has been detected in both experimental animal models of MS and within the CNS of affected MS patients. Furthermore, both IgG antibodies and complement activation can be detected in the demyelinating plaques and cortical gray matter lesions. We propose here that both immunoglobulins and complement play an active role in the neurodegenerative process of MS. We hypothesize that the increased CNS IgG antibodies form IgG aggregates and bind complement C1q with high affinity, activating the classical complement pathway. This results in neuronal cell damage, which leads to neurodegeneration and demyelination in MS.

## 1. Introduction

Multiple sclerosis (MS) is the most common inflammatory demyelinating disease of the central nervous system (CNS), affecting young adults in Northern temperate zones with profound demyelination and neurodegeneration leading to long-term disability. More than 2.8 million people worldwide live with MS with no definitive cure in sight [1]. Axonal loss is a key pathological feature of MS [2], and it is well established that irreversible loss of axons and neurons is the primary cause of most MS patients’ progressive neurological disability [3,4,5]. While it is clear that immunological abnormalities such as cerebrospinal fluid (CSF) oligoclonal bands and raised immunoglobulins are characteristic of MS, a key question is how the neurodegeneration typical of the disease is caused. We argue here that both immunoglobulins and complement play a significant role in the pathophysiology of MS [6] causing neurodegeneration, as the principal effector of an antibody-mediated immunological response is complement activation, and IgG in MS CSF is complement-fixing. We hypothesize that the increased CNS IgG antibodies form IgG aggregates and bind complement C1q with high affinity, causing neuronal cell damage, which leads to neurodegeneration (Figure 1). When referring to MS, we include all phenotypes of the disease, including primary progressive MS (PPMS), secondary progressive MS (SPMS), and relapsing and remitting MS (RRMS).

## 2. Neuron Damage and Neuronal Cell Loss Are the Primary Causes of MS Disease Progression and Disability

### 2.1. Grey Matter Pathology Correlates with Physical Disabilities and Cognitive Impairment in MS

MS has several destructive hallmarks, including acute damage to the neuroaxonal unit, progressive neuroaxonal degeneration, and brain atrophy, resulting in long-term disability. Pathological changes in the CNS gray matter, including the cortex, are central to the progression of MS [7]. Axonal injury correlates with demyelination [8], whereas gray matter pathology correlates with physical disabilities and cognitive impairment in MS [9,10,11,12]. We recently reported that plasma IgG aggregates in SPMS are elevated compared to both RRMS and PPMS [13], which caused significantly higher levels of neuronal apoptosis than in RRMS [14], supporting that neural cell apoptosis plays a vital role in MS pathogenesis [5]. The current hypothesis is predicated on the view that neurodegeneration is a leading cause of the disability seen in MS patients.

### 2.2. Demyelination and Neuron Loss May Occur Simultaneously in MS CNS

Demyelination of cerebral white matter is the pathological hallmark of MS. The classical notion is that demyelination is the consequence of inflammation, contributed by proinflammatory cytokines, CD8+ T cells, anti-myeline antibodies, and activated macrophages [14]. The disease process in MS causes the destruction of oligodendrocytes, which make myelin in the CNS, and this leads to demyelination [15]. Oligodendrocyte apoptosis was found in the new lesions of MS [16]. The pathological mechanism resulting in oligodendrocyte damage or death is not completely understood. Still, it seems highly likely there is a form of immune-mediated attack on these cells. Multiple factors have been shown to cause damage to oligodendrocytes, such as anti-myelin autoantibodies, cytotoxic T cells, and reactive oxygen species [17]. Demyelination is thought to drive neuronal degeneration and permanent neurological disability in individuals with MS [18]. However, oligodendrocyte and neuron damage resulting in neuronal cell loss may co-occur, with neuron loss being the leading event in the disease. Arguably, axonal damage may play a pivotal pathological role in MS, leading to secondary rather than primary demyelination. Recent studies suggest that cerebral white-matter demyelination and cortical neuronal degeneration can be independent events in the disease [18]. MS is conceptualized as a neurodegenerative rather than a pure neuroinflammatory disease with early involvement of axonal loss [2]. A recent study reported an inverse relationship between axon degeneration and demyelination [19].

### 2.3. Role of Complement in Animal Models of MS Neurodegeneration

Animal models of MS have yielded helpful information and insights into possible mechanisms of demyelination and neurodegeneration. An experimental allergic encephalomyelitis (EAE) model of MS showed that serum IgG antibodies from EAE rabbits cause demyelination [20]. Another earlier study showed that autoantibody depletion ameliorates disease in EAE [21]. An important finding in this area of complement-induced neurodegeneration was recently reported by Linzey et al. [22]. They studied two distinct murine models of MS, namely the proteolipid protein (PLP)-induced autoimmune EAE model (mimicking RRMS) and Theiler’s murine encephalomyelitis virus-induced demyelinating disease (mimicking progressive MS). Interestingly, it was found that the chronic-progressive disease form was more dependent on the classical complement pathway and protected the mice from acute relapses, whereas, by contrast, the relapsing model was more dependent on the alternative complement pathway. These results show that complement may play distinct roles at the various stages of MS.

## 3. Lack of Understanding of the Disease Mechanism Is a Critical Barrier to Effective Treatments in Progressive MS

Irreversible loss of axons and neurons is the primary cause of the progressive neurological decline that most MS patients endure [5]. Although current therapeutic strategies (>20 disease-modifying therapies) are beneficial during RRMS by modulating or suppressing immune function, treatment options for progressive MS are limited [9], and no effective therapies are available to prevent neuronal damage and slow disease disability. Therefore, a better understanding of the underlying mechanisms of neuroaxonal damage leading to neurodegeneration is a critical step toward finding improved treatment options.

Often, drugs prescribed for the treatment of MS act as disease-modifying therapies (DMTs), which function to reduce relapse frequencies during RRMS and lessen symptoms [23]. However, options for treating PPMS and SPMS remain limited due to an incomplete understanding of their pathologies. As the pathogenesis of MS continues to be elucidated, treatments increase in efficacy, with more recent treatments targeting the sphingosine 1-phosphate (S1P) signaling pathway or acting as monoclonal antibodies against B-lymphocyte antigen CD20 or interleukin-2 receptor alpha protein CD25 [24]. S1P DMTs, such as Siponimod, enter the central nervous system via the blood-brain barrier (BBB), binding to S1P receptors and functioning to slow disability and promote the regeneration of myelin [24]. Conversely, monoclonal antibodies such as Ocrelizumab induce complement-dependent cytotoxicity (CDC) and antibody-dependent cellular toxicity (ADCC) onto CD20-positive B cells. However limited, current treatments show short-term improvement in easing the progression of the disease, with a recent clinical trial finding disease progression to be 9.1% lower in Beta-1a interferon versus 13.6% lower in Ocrelizumab after 12 weeks [25]. Despite their short-term efficacies, current DMTs only act to postpone disease progression, reaching eventual plateaus where further treatments hold little value when compared to their risk of adverse events [26].

## 4. IgG Antibodies and Complement Activation Are Consistently Found in Demyelinating Plaques and Cortical Grey Matter Lesions

### 4.1. Increased Intrathecal Synthesis of IgG and Oligoclonal Bands Are the Most Characteristic Features of MS, and Evidence Supports the Pathological Role of IgG in MS

A consistent observation in MS is the presence of CSF oligoclonal bands, which occur in over 90% of cases [24]. Once present, the intrathecal IgG, mainly consisting of IgG1 and IgG3 [27], remains stable over time. Accumulating evidence supports the pathological role of CSF immunoglobulins. CSF OCBs were associated with increased levels of disease activity and disability, with the conversion from a clinically isolated syndrome (CIS) to early RRMS, along with more significant brain atrophy and increased disease activity [28,29,30,31,32,33,34,35]. Further, CSF in MS patients induced inflammatory demyelination and axonal damage in mice [36]. It has also been shown that some myelin-specific antibodies derived from MS CSF can cause both demyelination and complement-dependent cytotoxicity in mouse cerebellar oligodendrocytes [36]. Alcázar et al. reported that the soluble factors in primary progressive MS (PPMS) CSF can induce axonal damage and neuronal apoptosis [37]. Additionally, caspase inhibitors are shown to be protective against neuronal apoptosis induced by MS CSF [38]. Most recent studies demonstrated that CSF IgG in PPMS is pathogenic and causes motor disability and spinal cord pathology, including demyelination, impaired remyelination, reactive astrogliosis, and axonal damage [39]. These studies support the pathogenic effects of CSF IgG in MS.

### 4.2. Evidence Supports the Pathological Role of Blood IgG Antibodies in MS

Lisak et al. reported that MS plasma B cells secreted a factor or factors that induced apoptosis in cultured rat neurons [40]. Our recent report confirms the presence of IgG aggregate-mediated complement activation and cytotoxicity in astrocytes and neurons [41]. The extensive evidence for the role of complement in MS has recently been reviewed in detail by Saez-Calveras and Stuve [42]. These results support the early studies that most MS sera (>80%) caused demyelination (with natural complement) in newborn rat cerebellum [43]. It is also of relevance that we have recently shown that MS plasma contains IgG aggregates that induce complement-dependent neuronal cytotoxicity [41]. These MS plasma IgG aggregates may also function as biomarkers for the disease [41].

### 4.3. IgG Effector Functions

While extensive research has failed to identify the antigen specificity of the IgG antibodies in MS, there is nevertheless evidence to support IgG effector functions and their role in disease pathogenesis. The IgG Fc domain mediates a wide range of effector functions, including anti-body-dependent cellular cytotoxicity (ADCC) and complement-dependent cytotoxicity (CDC). For CDC, binding of complement component 1q (C1q) triggers the activation of the complement cascade and leads to the formation of the membrane attack complex (MAC), creating pores in the cell membrane and causing the lysis of target cells [44].

### 4.4. A Brief Introduction of Complement Activation

The complement system is a unique bridge between innate and adaptive immunity. It contributes to the removal of invading pathogens and dead or dying cells [45]. Complement activation plays a critical role in synaptic pruning during brain development [46]. It hails from three unique methods of activation: the classical pathway, the alternative pathway, and the lectin pathway. The classical pathway starts with complement C1, which consists of serine proteases C1r and C1s and the hexameric molecule C1q. C1q binds to the Fc regions of IgGs, and IgM is bound to antigens, which triggers the serine proteases to cleave C2 and C4 into fragments, resulting in the formation of C3 convertase [47,48]. The lectin pathway involves soluble pattern-recognition molecules, most notably mannose-binding lectin (MBL), which binds to carbohydrate signatures on bacteria or fungi, activating MASP-2, and forming C3 convertase [49]. Finally, the alternative pathway involves the hydrolysis of C3 into an initial form of C3 convertase, C3 (H_2_O) Bb, by Factor D. Further cleavage in the alternate pathway involving Factor B leads to the formation of C3 convertase and C5 convertase [47]. Though originating from different means of activation, the complement system always results in the production of anaphylatoxins such as C3a, opsonins such as C3b, and the C5b-9 membrane attack complex (MAC) [47]. The insertion of MAC causes necrosis of the cells, resulting in ruptures of the membrane and cell death. Ninomiya and Sims reported that inhibitor CD59 binds complement proteins C8 and C9 at the membrane to prevent insertion and polymerization of MAC pores [50].

### 4.5. IgG and Complement in MS Brain Lesions

Excessive amounts of IgG antibodies were reported in early studies of MS autopsy active lesions in both free/soluble and tissue-bound/particulate forms [51,52,53]. The IgG antibodies purified from corresponding soluble and particulate samples showed OCBs [53]. Over 20 times more IgG was extracted from MS plaques than control brain [54]. In the landmark paper by Lucchinetti et al., antibodies and complements were found extensively in pattern II, whereas patterns III and IV were suggestive of oligodendrocyte damage/loss [55]. The lack of detection of IgG antibodies in the latter patterns may indicate the limitation of the detection method, as only immunohistochemistry was used.

The co-localization of IgG antibodies, complement, and Fc gamma receptors (FcγR) in the active lesions was reported, suggesting the pathological role of these antibodies in the early stages of MS [56]. Furthermore, complement activation in cortical gray matter lesions of brain tissues in patients with progressive MS [57] implies that antibodies and complements may contribute to the pathological mechanism underlying the disease’s irreversible progression. The dual roles of the C5b-9 complement complex in demyelination have been discussed extensively in early studies [58]. The activation of C5b-9 promotes demyelination, and the sublytic C5b-9 can protect oligodendrocytes from programmed cell death.

### 4.6. Complement Activation in MS CSF

Early studies reported a highly significant reduction of the terminal component of complement (C9) concentration in MS CSF [59]. Reduced level of CSF C9 in MS implies complement activation resulting in C9 consumption for the formation of membrane attack complexes, leading to myelin damage and neuronal cell membrane damage and causing a more widespread but reversible loss of function. More recently, it was shown that complement activation in the CSF in clinically isolated syndrome and early stages of relapsing-remitting MS [60]. Significantly, they found that CSF C1q and CSF C3a correlated with neuroinflammatory markers and neurofilament light chain levels; CSF C3a correlated with disease activity and progression. A very recent study reported that complement activation is associated with disease severity in MS [61]. They measured complement components and complement activation products in the CSF and plasma in 112 patients with clinically isolated syndrome (CIS), 127 patients with MS, and 75 CNS controls. They showed that C3a and C4a in MS CSF were associated with increased EDSS scores. The CNS compartmentalized activation of the classical and alternative pathways of complement support that complement activation contributes to MS pathology and disease severity.

## 5. Complement and IgG in Other Autoimmune and Neurodegenerative Diseases

### 5.1. Complement and IgG in Other Autoimmune Diseases

Complement activation is also a feature of other autoimmune diseases. For example, in systemic lupus erythematosus (SLE) with vasculitis, the levels of serum complement (C3, C4) were reduced [62], with complement levels being known to be lowered in SLE [63]. Complement activation was also reported in patients with nephrotic glomerular diseases [64]. In Myasthenia gravis, which is a prototype autoimmune disease, complement activation can be detected on the muscle membrane surface [63]. Given the known involvement of complement pathways in autoimmune diseases, anti-complement therapies have been considered and developed as therapeutic measures. Whether such therapies may be given to MS patients is a more controversial issue.

Myelin oligodendrocyte glycoprotein (MOG) antibody immunoglobulin G (IgG)-associated disease (MOGAD) has clinical and pathophysiological features-like but distinct from those of aquaporin-4 antibody (AQP4-IgG)-positive neuromyelitis optica spectrum disorders (AQP4-NMOSD). Both MOG-IgG and AQP4-IgG can activate the complement system [65,66]. A recent study showed higher complement activation and C3 degradation in MOGAD sera compared to NMOSD; in contrast, higher levels of soluble C5b-9 (sC5b-9) in NMOSD [67]. These studies further support the complement and IgG in the pathological roles of these disorders.

### 5.2. Complement and IgG in Other Neurodegenerative Diseases

The complement system is integral for protection against pathogens and for normal development of the CNS [68]. Several cell types in the brain have been implicated in the complement system, such as microglia and astrocytes [69]. Astrocytes are critical for many functions in the brain, such as clearance of cellular debris and aberrant proteins, maintaining the blood-brain barrier, and recycling of neurotransmitters [70]. Microglia are the resident immune cells in the brain. They continuously monitor for damaged cells and pathogenic invaders. When they locate something potentially harmful, they become activated, breaking down and engulfing foreign substances. They are also critical in many homeostatic functions, such as neurogenesis, astrogenesis, and maintaining synapses [71]. Astrocytes and microglia work in a concerted effort to maintain homeostasis in the brain and mediate inflammation [69]. Both microglia and astrocytes are known to play a role in cytokine production as well as secretion of both cytokines and complement proteins [72].

The complement pathway has been implicated in several neurodegenerative diseases such as Alzheimer’s disease (AD), Parkinson’s disease (PD), and Amyotrophic lateral sclerosis (ALS). Complement has a vital role in eliminating pathogens, clearing debris, and maintaining synaptic function in the central nervous system (CNS) [73]. However, when this highly regulated system begins to fail, it may exacerbate the progression of neurodegenerative diseases [69]. Several glial cell types have been implicated in neurodegeneration. Astrocytes and microglia are all necessary for normal functioning of the CNS, but when they become overactive in a diseased or aging state this may result in an increase of neuropathology in a degenerative state [73].

When astrocytes become reactive, they release C3. This results in the cleavage of C3 into its subunits, C3a and C3b. Microglia expresses the C3a receptor (C3aR). When bound by this ligand, microglia become activated, initiating the inflammatory response and phagocytosis, which is a potential mechanism of early synaptic loss in AD, resulting in memory impairments in mouse models [74]. Complement proteins in the CSF of AD patients have shown increased levels of C1q, C3d, and C4d [75,76]. Similarly, it has been shown in PD there is an increase of complement factors C3d, C4d, C7, and C9 in proximity to Lewy Bodies [77]. In mice, it was also shown that a C3 receptor (C3R) knockout was protected from the loss of dopaminergic neurons and motor impairments, which suggests that complement may contribute to the worsening of symptoms in PD [78].

### 5.3. The Involvement of Complement in Astrocytes and Microglia in MS

Complement C3 on microglial clusters in MS was found in the chronic stage of MS [79]. MS microglia nodules are associated with IgG transcription, complement activation, and MAC formation [80]. These studies provide evidence that complement activation in microglia may provide novel mechanisms of MS pathology.

## 6. Virus Infection and Complement Activation

Since a virus is thought by many to be involved in MS pathogenesis [27,81], it is relevant to consider whether complement activation occurs during viral infection. The role of complement in various viral infections has been reviewed extensively by Stoermer and Morrison [82]. Some viruses, such as Human Immunodeficiency virus (HIV) and cytomegalovirus (CMV), have developed strategies to evade the complement system by recruiting host complement regulatory proteins into their virions [82], thereby indicating that complement plays a role in virus-induced pathology. On the other hand, complement pathways can also protect the host against viral infection by mediating a degree of protection from damage and pathology induced by the virus. This protection against virus-induced disease can also occur via complement-mediated enhancement of B lymphocyte responses. This complement can either produce or protect against viral infection.

## 7. Hypothesis: IgG and Complement as Critical Players in Neurodegeneration in MS

### 7.1. Demyelination by Autoantibodies Requires Complement Activation without Fc Receptor Activation

Complement activation was found in both the early and later progressive phases of MS [83,84]. The ubiquitous occurrence of IgG and C1q staining in MS lesions indicates the dominant role of IgG antibodies and the classical complement pathway [56,85]. Antibody-antigen immunocomplexes were detected in foamy macrophages in the active lesion areas of MS [86]; demyelination by autoantibodies was found to be dependent on complement activation without Fc receptor activation [87]. A recent study demonstrated a strong association between neurodegeneration and local complement activation in progressive MS [88]. These studies provide further support for IgG-mediated complement activation in MS disease pathophysiology.

### 7.2. MS IgG and IgG Complexes Bind to the Surface of Neurons and Myelin

The specific antigens of the IgG within OCB in MS have remained to be identified [27]. Prominent antigen candidates include myelin proteins and viruses [81]. OCB antibodies were shown to be directed against ubiquitous intracellular antigens [89]. We previously reported MS OCB target patient-specific peptide antigens [81], and EBV epitopes were shown to be reactive to OCB [90]. It was reported that most MS serum antibodies are reactive to brain components [91].

Our study that MS plasma IgG aggregates/immune complexes induced apoptosis in neurons [41] suggests that the IgG aggregates may bind to the surfaces of neurons, which is consistent with the finding that MS had significantly higher levels of serum IgG bindings to the cell surfaces of neurons and oligodendrocytes [92]. Furthermore, IgG aggregates and immune complexes have been shown to bind myelin in MS [93]. Recent studies showed that MS CSF IgG binds to the human oligodendroglioma cell line, discriminating MS from controls with 96% specificity [92].

### 7.3. C1q Has a Low Affinity to Monomeric IgG but Binds to IgG Hexamers with High Avidity for Efficient Complement Activation

Complement activation is one of the most important biological functions of IgG. IgG immune complexes can activate all three pathways of the complement system, resulting in the generation of C3 and C5 cleavage products, activating a panel of different complement receptors on innate and adaptive immune cells [47]. The interaction of antibodies with specific antigens forms immune complexes/IgG aggregates. C1q has a very low affinity for monomeric IgG but avidity for IgG hexamers, which promotes complement activation [94], resulting in tissue damage and dynamic systemic activation of complement [57]. Indeed, our recent study demonstrated that MS IgG aggregate-induced neuronal cytotoxicity is complement-dependent; a co-localization of IgG and C3b was detected in neurons after incubation with MS plasma IgG aggregates [41]. We showed the staining of C5b-9 in the neurons, but we do not know whether C5b-9 is lytic or sublytic. Further investigations are needed to elucidate the mechanism.

### 7.4. Evidence Supporting the Presence of IgG Aggregates in MS CNS

Both IgG1 and IgG3 subclasses were reported to be present in the same OCB band, suggesting OCB consisted of heterogeneous antibodies [95]. We reported previously that there is no correlation between CSF IgG concentrations and the number of OCBs, suggesting the presence of IgG aggregates in the CSF [96]. In addition, significantly higher amounts of bound IgG were eluted from the MS brain using high- or low-pH buffers [51]. The antibody-antigen immune complexes were detected in foamy macrophages in the active lesion areas [97]. Furthermore, Mehta et al. [51] reported the presence of IgG immune complexes in MS brain tissues of white and gray matter. These studies support the presence of IgG aggregates in MS.

### 7.5. Potential Mechanism of IgG-Mediated, Complement-Dependent Neurodegeneration in MS

Neuron-specific activation of necroptosis was found in the cortical gray matter of MS [98]. Neuronal and axonal degeneration in MS was thought to be initiated by acute inflammation and subsequently driven by chronically smoldering, diffuse parenchymal myeloid, and meningeal lymphocytic inflammation. Oxidative stress, mitochondrial injury, and subsequent ion channel dysfunction secondary to chronic inflammation seem to have a constant impact on neurons and axons, leading to their demise during progressive MS [99].

It should be appreciated, however, that several distinct factors, in addition to complement and IgGs, may lead to the neurodegeneration seen in MS, and they are certainly not mutually exclusive. This issue has recently been discussed in detail by Morgan et al. [100] who pointed out that complement-induced MS pathology may be different in white versus gray matter in MS, that it is not known how important serum-derived as opposed to CNS-derived complement is, and that the specific complement pathways of importance remain unknown. However, both C1q and C3 may play a role, and different pathogenic mechanisms may operate at the various stages of MS, e.g., RRMS and SPMS. Morgan et al. [100] provided evidence for two fundamental neurodegenerations induced by complement pathways in MS brains. The first was called “outside-in”, and the other was called the “inside-out” paradigm. In the former case, complement components enter the CNS via a breached Blood-brain barrier (BBB). They are activated either by the local presence of antigen/antibody complexes or else by “altered self” myelin epitopes. Local phagocytes and microglia then engulf myelin, which is attached to complement, resulting in demyelination and neurodegeneration. There is abundant experimental evidence to support this mechanism. By contrast, in the “inside-out” model, a so-far unknown key cellular degenerative event triggers myelin alterations and degeneration, which then allows very antigenic myelin epitopes to be exposed, with the complement pathways then acting as a secondary mechanism whereby they contribute to inflammatory responses followed by the tagging and phagocytosis of myelin [100]. Both scenarios are highly imaginative and creative, but it is yet unknown which, if either, operates in MS neurodegenerative processes. We hypothesize that both IgG and complement are key drivers of the neurodegeneration seen in MS. However, the actual mechanism(s) involved remains speculative if this is the case.

MS CNS is the target of the pathological immune responses affecting every part of the brain [101,102], characterized by the loss of neurons and oligodendrocytes, resulting in demyelination and neurodegeneration [103,104]. Gray matter pathology plays a central role in disease progression [105]. Meningeal B-cell follicles in SPMS are associated with the early onset of disease and severe cortical pathology [106], suggesting that soluble factors (IgG aggregates?) diffusing from these structures has a pathogenic role in neurons. We hypothesize that in MS CNS, the compartmentalized IgG antibodies form aggregates and produce complement-dependent cytotoxicity in neural cells.

## 8. Conclusions

The pathogenesis of demyelination and neurodegeneration in MS is not well understood. It is established that MS is characterized by the presence of oligoclonal bands and raised IgG. In addition, there is good evidence that complement activation is present in MS, both in histopathology of the MS brain lesions, in CSF, and in experimental studies in animal models of the disease. A key pathological feature of the MS lesion is axonal loss, which not only correlates with demyelination but is also likely to be the cause of the neurodegeneration seen as the disease progresses. It is hypothesized here that a combination of IgG and complement plays a significant role in the neurodegenerative process of MS. If this proves to be the case, then consideration should be given to developing effective therapies based on antagonists or antibodies to complement and its biochemical pathways. It is pertinent to consider future perspectives and directions. Further research is required to confirm and delineate a pathogenic role for IgG and complement in demyelination and neuronal cytotoxicity rather than just being epiphenomena. At this stage, it would be premature to consider the therapeutic options for blocking the actions of these two components in some way; nevertheless, this remains a potential treatment possibility in the future. It is also possible that clinical trials with other drugs or agents may add weight, or otherwise, to this notion.

## Figures and Tables

**Figure 1 biomolecules-14-01210-f001:**
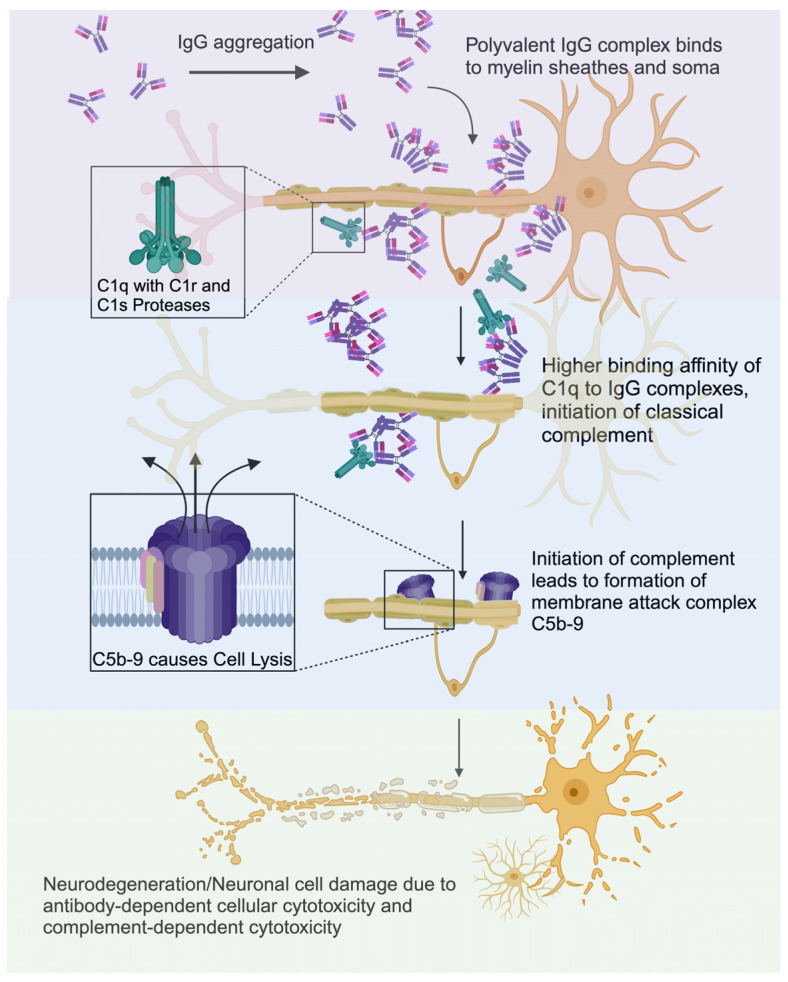
C1q exhibits increased binding to aggregated IgGs, which attach to the myelin and soma, leading to the C5b-9 membrane attack complex to form and lyse host-originating cells. Created with BioRender.com.

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
