# Peer review of "Immunoglobulin G and Complement as Major Players in the Neurodegeneration of Multiple Sclerosis"

_biomolecules, 2024, doi:10.3390/biom14101210_

Round 1

Reviewer 1 Report

Comments and Suggestions for Authors

Peter GE Kennedy et al., presented a review article titled, “Immunoglobulin G and Complement as Major Players in the Neurodegeneration of Multiple Sclerosis”. The authors aimed to correlate the implications of increased CNS IgG antibodies leading to IgG aggregates and binding to complement C1q in activating the classical complement pathway. The authors hypothesized the significant role of IgG and Complement in neuronal cell damage, subsequently leading to neurodegeneration and demyelination in MS.

The manuscript discussed the neuron damage and neuronal cell loss are the primary causes of MS disease, the disease mechanism, IgG antibodies and complement activation, complement and IgG in other autoimmune and neurodegenerative diseases, virus infection - complement activation and discussed the undertaken hypothesis with suitable literature.

The manuscript was presented clearly and the literature covers all the basic information in the introduction section.

Figure 1 is appropriate and presented, clearly.

References covered recent literature and are adequate.

However, the below-mentioned minor revision is required before the acceptance of the manuscript:-

1.     Plagiarism is detected in sections 2.2.; 4.3.; 4.5.; 4.6.; 7.1.; 7.4; and 1st para of 7.5 Hence, the authors need to address this issue (report file is attached).

2.     If it is available, the authors need to provide a Table consisting of clinical trial candidates, or currently applicable therapeutic approaches about the topic of the review.

3.     Where possible, authors should cite recent literature.

4.     In conclusion, authors need to include the future perspectives.

Author Response

R1

We appreciate the reviewer’s comments on the manuscript's merits, including “The manuscript was presented clearly, and the literature covers all the basic information in the introduction section; Figure 1 is appropriate and presented clearly; references covered recent literature and are adequate.”

  1. Plagiarism is detected in sections 2.2.; 4.3.; 4.5.; 4.6.; 7.1.; 7.4; and 1stpara of 7.5. Hence, the authors need to address this issue.

We are grateful for the reviewers’ concerns regarding possible plagiarism, which we take very seriously. Though the text is fully referenced, there is some repetition of published sentences. We have now either reworded or deleted sentences of concern. We also ran a computer plagiarism detection program on the revised text, and no issues were detected. We believe this issue is now entirely resolved.

  1. If it is available, the authors need to provide a Table consisting of clinical trial candidates, or currently applicable therapeutic approaches about the topic of the review.

Extensive reviews regarding clinical trials of B cell depletion therapies and complement inhibitors have been discussed elsewhere (Greenfield and Hauser, 2018; Nil Saez-Calveras and Olaf Stuve, 2022). We do not feel providing a similar table in this manuscript is necessary.

  1. Where possible, authors should cite recent literature.

We feel that the manuscript covered a representative sample of recent literature. Indeed, this reviewer praised the “references covered recent literature and are adequate.” 

  1. In conclusion, authors need to include the future perspectives.

Thank you for the suggestion! We included future perspectives in the conclusion section of the revised manuscript.

Reviewer 2 Report

Comments and Suggestions for Authors

Critique

1.    The manuscript needs to address the beneficial role of complement activation and C5-9 in demyelination. There is significant work especially related to oligodendrocytes and sublytic C5b-9, which was not mentioned by the authors,

2.    The authors need also to discuss demyelination not as secondary to oligodendrocyte cell death, but also as a direct attack on myelin which wraps around the axons (paragraph 2,2 on page 3).

3.    Activation of C5b-9 on cells surface is prevented by surface complement inhibitors like CD59 and others. This also needs to be discussed in this review. Manny time C5b-9 insertion in the cell membranes is secondary, the so called “bystander cell lysis”.

4.    The landmark paper of Lucchinetti et al from 2000 (Ann Neurol. 2000 Jun;47(6):707-17.) on MS pathology heterogeneity should be discussed in the context of immunoglobulins and complement activation. In addition, work by Prines (Ann Neurol. 2012 Jul;72(1):18-31. doi: 10.1002/ana.23634.and Mult Scler. 2018 Apr;24(5):610-622. doi: 10.1177/1352458517706037.) should be discussed.

5.    Lane 4 from introduction: “MS with few options for intervention”. This is not accurate as we have more than 20 DMTs.

6.    Page 3, paragraph 3. Please delete references to Daclizumab as this medication had many side effects and is not on the marked for many years.

7.    Page 7, paragraph 7.3. Complement activation is one of the most important (not the most important).

8.    At the end of paragraph 7.3: please discuss colocalization of immunoglobulins with C5b-9. Is this C5b-9 lytic or sublytic?

9.    In paragraph 9, before Morgan et at please also discuss C5b-9 in MS acute vs chronic lesions. Also, white matter C5b-9 deposits vs gray matter deposits. Are there any differences?

Author Response

R2

  1. The manuscript needs to address the beneficial role of complement activation and C5-9 in demyelination. There is significant work especially related to oligodendrocytes and sublytic C5b-9, which was not mentioned by the authors.

Thank you for the excellent point. In the revised manuscript, we described the beneficial role of complement activation in eliminating invasive pathogens and synapse pruning during development (section 4.5). We mentioned the extensive review of the dual role of C5b9 in demyelination (section 4.5).

  1. The authors need also to discuss demyelination not as secondary to oligodendrocyte cell death, but also as a direct attack on myelin which wraps around the axons (paragraph 2,2 on page 3).

In the revised manuscript, we discussed demyelination as a direct attack on myelin (section 2.2).

  1. Activation of C5b-9 on cells surface is prevented by surface complement inhibitors like CD59 and others. This also needs to be discussed in this review. Manny time C5b-9 insertion in the cell membranes is secondary, the so called “bystander cell lysis”.

We included the discussion of the inhibitor CD59 for MAC in the revised manuscript (section 4.5).

  1. The landmark paper of Lucchinetti et al from 2000 (Ann Neurol. 2000 Jun;47(6):707-17.) on MS pathology heterogeneity should be discussed in the context of immunoglobulins and complement activation. In addition, work by Prines (Ann Neurol. 2012 Jul;72(1):18-31. doi: 10.1002/ana.23634.and Mult Scler. 2018 Apr;24(5):610-622. doi: 10.1177/1352458517706037.) should be discussed.

Findings from Lucchinetti et al. (2000) and Prineas et al. (2012 & 2018) were cited and discussed in the revised manuscript (sections 4.6 & 7.2).

  1. Lane 4 from introduction: “MS with few options for intervention”. This is not accurate as we have more than 20 DMTs.

The sentence was deleted.

  1. Page 3, paragraph 3. Please delete references to Daclizumab as this medication had many side effects and is not on the marked for many years.

This sentence was deleted.

  1. Page 7, paragraph 7.3. Complement activation is one of the most important (not the most important).

Revised.

  1. At the end of paragraph 7.3: please discuss colocalization of immunoglobulins with C5b-9. Is this C5b-9 lytic or sublytic?

We included the following in section 7.3. We showed the staining of C5b9 in the neurons, but we do not know whether C5b9 is lytic or sublytic. Further investigations are needed to elucidate the mechanism.

  1. In paragraph 9, before Morgan et at please also discuss C5b-9 in MS acute vs chronic lesions. Also, white matter C5b-9 deposits vs gray matter deposits. Are there any differences?

Although the role of C5b9 warrants an extensive discussion of its role as lytic or sublytic in neuronal cells, acute vs. chronic lesions, and white vs. grey matter, we do not feel we have the expertise and space to add such a discussion to the current manuscript.

Reviewer 3 Report

Comments and Suggestions for Authors

Below is a minor suggestion that might help with clarity.  In addition, a few potential additional references are listed for consideration.

1. It might be useful to include a table that connects different stages of MS based on biological markers with key aspects of the disease stage, including appropriate references.  This would serve as a quick reference for navigating what is known about the overall disease.

2. Additional potential references for inclusion/discussion:

PMID: 37903797

PMID: 32400866

PMID: 36280798

PMID: 38513664

Author Response

R3

Below is a minor suggestion that might help with clarity.  In addition, a few potential additional references are listed for consideration.

  1. It might be useful to include a table that connects different stages of MS based on biological markers with key aspects of the disease stage, including appropriate references.  This would serve as a quick reference for navigating what is known about the overall disease.

Thank you.  We included new references on MS biomarkers with disease stage/phenotype in the revised manuscript. We believe this would serve as a quick reference for navigating what is known about the disease. There are excellent reviews on biological markers for MS (https://www.ncbi.nlm.nih.gov/pmc/articles/PMC6396336/, https://www.sciencedirect.com/science/article/pii/S0009898123002735#s0035). We reported last year that plasma IgG aggregates as a high sensitivity biomarkers to differentiate secondary progressive MS from RRMS (Zhou 2023, https://www.sciencedirect.com/science/article/pii/S1521661623005648#:~:text=Using%20ELISA%2C%20we%20report%20herein,by%20detection%20of%20IgG%20subclass.). Recent studies show that serum GFAP and NfL Levels Differentiate Subsequent Progression and Disease Activity in Patients With Progressive MS (https://www.neurology.org/doi/10.1212/NXI.0000000000200052). However, it is repetitive and beyond the scope of this manuscript to include such a table in the revised manuscript.

  1. Additional potential references for inclusion/discussion:

Thank you for the excellent references. We included PMID: 37903797 in section 2.2. “ A recent study reported an inverse relationship between axon degeneration and demyelination.

We did not include the remaining three references as these papers are about mechanisms of MS white matter legions, which are not the focus of the current manuscript.

Reviewer 4 Report

Comments and Suggestions for Authors

In this review, the authors introduce the hypothesis that IgG and complement activity are involved in the pathological progression of MS, but the logic used to prove this hypothesis is not adequate.

Fundamental changes are needed, particularly in the following respects.

1.     Clearly separate the discussion of the role of IgG and complement into serum and intrathecal findings.

OCB is a condition that produces selective intrathecal antibodies. On the other hand, studies such as reference 41 examined the involvement of serum-derived IgG and complement. They are completely different pathologies and it is inappropriate to discuss them identically (e.g. SECTION 4.1)

2.     When comparing the pathophysiology of complement activity in MS to complement activity in other diseases, assess appropriately what the purpose of the comparison is and its validity.

Why did the authors introduce the pathogenesis of SLE, nephrotic glomerular disease and MG, instead of considering diseases with clinical similarities such as NMOSD and MOGAD?

(e.g. SECTION 5.1)

The authors describe the involvement of astrocytes and microglia in neurodegenerative diseases, but make no mention of their role in MS. (e.g. SECTION 5.2)

Intracellular protein aggregation in neurodegenerative diseases and tissue deposition of immune complexes are completely different pathologies (e.g. SECTION 7.4).

3.     Describe appropriately which data on IgG and complement are relevant to which pathology (relapse, severity, progression) of MS.

It is not reasonable to use data on CIS and RRMS to conclude that they are involved in disease progression in MS. If one wants to point out the involvement of progression, data on PPMS/SPMS or PIRA (progression independent of relapse activity) and complement should be presented (e.g. SECTION 4.6).

Author Response

R4

In this review, the authors introduce the hypothesis that IgG and complement activity are involved in the pathological progression of MS, but the logic used to prove this hypothesis is not adequate.

Fundamental changes are needed, particularly in the following respects.

  1. Clearly separate the discussion of the role of IgG and complement into serum and intrathecal findings.

 OCB is a condition that produces selective intrathecal antibodies. On the other hand, studies such as reference 41 examined the involvement of serum-derived IgG and complement. They are completely different pathologies and it is inappropriate to discuss them identically (e.g. SECTION 4.1).

Thank you for your suggestions and comments. We added a new subsection (4.2) providing evidence of serum IgG antibodies in the pathogenesis of MS.

      2. When comparing the pathophysiology of complement activity in MS to complement activity in other diseases, assess appropriately what the purpose of the comparison is and its validity. Why did the authors introduce the pathogenesis of SLE, nephrotic glomerular disease and MG, instead of considering diseases with clinical similarities such as NMOSD and MOGAD?

The goal of this manuscript is to provide evidence supporting the pathological roles of IgG and complement in MS. To support our hypothesis, we wanted to provide findings on IgG and complement activation from well-studied autoimmune diseases such as SLE, nephrotic glomerular disease, and MG. In the revised manuscript, we added findings on complement activation in NMOSD and MOGAD (section 5.1).

 The authors describe the involvement of astrocytes and microglia in neurodegenerative diseases, but make no mention of their role in MS. (e.g. SECTION 5.2).

In the revised manuscript, we added a new subsection (5.3), providing evidence of complement activation in MS microglia.

Intracellular protein aggregation in neurodegenerative diseases and tissue deposition of immune complexes are completely different pathologies (e.g. SECTION 7.4).

We agree. Minimal studies have been done on protein aggregates in MS. We present data on protein aggregation in neurodegenerative diseases and tissue deposition of immune complexes to support our hypothesis that IgG aggregates may be present in MS and cause complement-dependent cytotoxicity.

  1. Describe appropriately which data on IgG and complement are relevant to which pathology (relapse, severity, progression) of MS.

It is not reasonable to use data on CIS and RRMS to conclude that they are involved in disease progression in MS. If one wants to point out the involvement of progression, data on PPMS/SPMS or PIRA (progression independent of relapse activity) and complement should be presented (e.g. SECTION 4.6).

We apologize for the oversight. The revised sentence is “…complement activation contributes to MS pathology and disease severity.” (section 4.7).

Round 2

Reviewer 2 Report

Comments and Suggestions for Authors

Critique

1.Paragraph 4.5. Please correct/finish the sentence bellow:

It contributes to the removal of invading pathogens and dead or dying [47].

2. Paragraph 4.5, last sentence (see below). Role of CD59 as inhibitor of C5b-9 assembly was delineated many years ago in the 1990s. Please remove recently.

“It was reported recently that inhibitor CD59 binds”

3. Paragraph 4.6 second sentence “soluble and particulate samples displayed OCBs”

Please clarify what you mean by “displaying OCB”.

4. Last 3 lanes of page 4, please replace C5b9 with C5b-9. There are also many other instances in the manuscript were C5b-9 is misspelled.

5. Paragraph 5.1 last sentence

“higher levels of C5b9 in NMOSD”

Are these higher levels higher levels of C5b-9 in serum? In that case this complex is called sC5b-9 (soluble C5b-9) which is cytolytically inactive and needs to be differentiated foam C5b-9 or MAC which inserts in the cell membrane.  Please change to sC5b-9.

Author Response

1.Paragraph 4.5. Please correct/finish the sentence bellow:

It contributes to the removal of invading pathogens and dead or dying [47].

We made the correction by adding “cells” to the end of the mentioned sentence; thank you.

  1. Paragraph 4.5, last sentence (see below). Role of CD59 as inhibitor of C5b-9 assembly was delineated many years ago in the 1990s. Please remove recently.

We revised the sentence and added the suggested reference. Thank you.

  1. Paragraph 4.6 second sentence “soluble and particulate samples displayed OCBs”

Please clarify what you mean by “displaying OCB”.

“Display” is synonymous with “show” which the referenced study used to communicate the analysis and detection of the oligoclonal IgG bands (OCBs) in the samples described. In the revised manuscript, we replaced the “display” with “show” to make the sentence clearer.

  1. Last 3 lanes of page 4, please replace C5b9 with C5b-9. There are also many other instances in the manuscript were C5b-9 is misspelled.

In the revised manuscript, we replaced C5b9 with C5b-9. Thank you.

  1. Paragraph 5.1 last sentence

“higher levels of C5b9 in NMOSD”

Are these higher levels higher levels of C5b-9 in serum? In that case this complex is called sC5b-9 (soluble C5b-9) which is cytolytically inactive and needs to be differentiated foam C5b-9 or MAC which inserts in the cell membrane.  Please change to sC5b-9.

We have reviewed the study and agreed with the reviewer. We greatly appreciate your expertise and critical review. We added “soluble C5b-9 (sC5b-9) in the revised manuscript for clarification. Thank you.

Reviewer 4 Report

Comments and Suggestions for Authors

There are still some points that need to be corrected in the following areas.

4.1. Increased intrathecal synthesis of IgG and oligoclonal bands are the most characteristic features of MS, and evidence supports the pathological role of IgG in MS.

“Our preliminary data showed that in MS plasma, there was a strong correlation between IgG antibodies in the aggregates and neuronal apoptosis.”

Data without citations should not be presented, and it is inappropriate to discuss data regarding plasma in the section on “Increased intrathecal synthesis of IgG and OCB”.

4.2. Evidence supports the pathological role of blood IgG antibodies in MS

In this section, the authors describe IgG-independent complement activation, but this does not support the pathological role of blood IgG antibodies in MS. The authors should clearly distinguish between those related to IgG itself, IgG-derived complement activity (classical pathway activation), and IgG-independent complement activity in MS.

4.3. Reduced levels of IgG induced by disease-modifying therapies in MS.

The phenomenon of a decrease in IgG should not simply be interpreted as a therapeutic effect. In general, the decrease in IgG/IgM associated with Fingolimod and B-cell depleting therapies is recognized as a side effect. If the authors' hypothesis is correct, it is necessary to present data showing that the decrease in IgG itself is related to prognosis. For example, is there any evidence that the more intense the decrease in IgG, the better the prognosis with the same treatment? If such data is not available, this section is not necessary.

4.6. IgG and complement in MS brain lesions.

The authors state that “the landmark paper by Lucchinetti et al. found antibodies and complements in Pattern I and Pattern II”, but this is incorrect. Only Pattern II involves complement.

7.4. Protein aggregates are common in other neurodegenerative diseases.

The description of protein aggregation has not been corrected. Do the authors think that protein aggregation in MS, accumulating in cells like α-synuclein in PD and TDP-43 in ALS, is involved in the progression of MS? This is completely different from the pathology that the authors are hypothesizing, and does not support their pathology. This section should be deleted because it gives the reader a false impression.

Figure1

The authors have proposed the hypothesis that the classical pathway is activated by the activation of C1q after IgG aggregates, but there is little evidence to support MAC formation after that. The classical pathway has various other functions (opsonization, chemotaxis), so it is preferable to describe multiple possibilities.

Author Response

4.1. Increased intrathecal synthesis of IgG and oligoclonal bands are the most characteristic features of MS, and evidence supports the pathological role of IgG in MS.

“Our preliminary data showed that in MS plasma, there was a strong correlation between IgG antibodies in the aggregates and neuronal apoptosis.”

Data without citations should not be presented, and it is inappropriate to discuss data regarding plasma in the section on “Increased intrathecal synthesis of IgG and OCB”.

We agree. This sentence was deleted. Thank you.

4.2. Evidence supports the pathological role of blood IgG antibodies in MS

In this section, the authors describe “IgG-independent complement activation”, but this does not support the pathological role of blood IgG antibodies in MS. The authors should clearly distinguish between those related to IgG itself, IgG-derived complement activity (classical pathway activation), and IgG-independent complement activity in MS.

In the revised manuscript, we deleted the part regarding “IgG-independent complement activation”.

4.3. Reduced levels of IgG induced by disease-modifying therapies in MS.

The phenomenon of a decrease in IgG should not simply be interpreted as a therapeutic effect. In general, the decrease in IgG/IgM associated with Fingolimod and B-cell depleting therapies is recognized as a side effect. If the authors' hypothesis is correct, it is necessary to present data showing that the decrease in IgG itself is related to prognosis. For example, is there any evidence that the more intense the decrease in IgG, the better the prognosis with the same treatment? If such data is not available, this section is not necessary.

We agree. This section was deleted.

4.6. IgG and complement in MS brain lesions.

The authors state that “the landmark paper by Lucchinetti et al. found antibodies and complements in Pattern I and Pattern II”, but this is incorrect. Only Pattern II involves complement.

This has been corrected, thank you.

7.4. Protein aggregates are common in other neurodegenerative diseases.

The description of protein aggregation has not been corrected. Do the authors think that protein aggregation in MS, accumulating in cells like α-synuclein in PD and TDP-43 in ALS, is involved in the progression of MS? This is completely different from the pathology that the authors are hypothesizing, and does not support their pathology. This section should be deleted because it gives the reader a false impression.

In the revised manuscript, we deleted the description of protein aggregates in other neurodegenerative diseases and changed the section title to “evidence supporting the presence of IgG aggregates in MS CNS.”

Figure1

The authors have proposed the hypothesis that the classical pathway is activated by the activation of C1q after IgG aggregates, but there is little evidence to support MAC formation after that. The classical pathway has various other functions (opsonization, chemotaxis), so it is preferable to describe multiple possibilities.

We are aware of the other functions of the complement pathway, but our central hypothesis involves activation of the classical pathway via aggregation of IgGs. It is well known that C1q shows low binding avidity to the Fc region of non-aggregated, non-hexameric IgGs, whereas studies involving hexameric IgG aggregates show an increase in C1q to IgG aggregate binding, thus leading to more efficient activation of the classical pathway and formation of the MAC, as discussed in the following paper:

Naseraldeen N, Michelis R, Barhoum M, et al. The Role of Alpha 2 Macroglobulin in IgG-Aggregation and Chronic Activation of the Complement System in Patients With Chronic Lymphocytic Leukemia. Front Immunol. 2021;11:603569. Published 2021 Feb 11. doi:10.3389/fimmu.2020.603569